# Preclinical Studies with Glioblastoma Brain Organoid Co-Cultures Show Efficient 5-ALA Photodynamic Therapy

**DOI:** 10.3390/cells12081125

**Published:** 2023-04-10

**Authors:** Leire Pedrosa, Carmen Bedia, Diouldé Diao, Alejandra Mosteiro, Abel Ferrés, Elisabetta Stanzani, Fina Martínez-Soler, Avelina Tortosa, Estela Pineda, Iban Aldecoa, Marc Centellas, Marta Muñoz-Tudurí, Ana Sevilla, Àngels Sierra, José Juan González Sánchez

**Affiliations:** 1Laboratory of Experimental Oncological Neurosurgery, Neurosurgery Service, Hospital Clinic de Barcelona—FCRB, 08036 Barcelona, Spain; 2Institute of Environmental Assessment and Water Research (IDAEA-CSIC), 08034 Barcelona, Spain; 3Department of Neurosurgery, Hospital Clínic de Barcelona, University of Barcelona, 08036 Barcelona, Spain; 4Laboratory of Pharmacology and Brain Pathology, IRCCS Humanitas Research Hospital, 20089 Milan, Italy; 5Apoptosis and Cancer Unit, Department of Basic Nursing, IDIBELL, Faculty of Medicine and Health Sciences, University of Barcelona, 08907 L’Hospitalet del Llobregat, Spain; 6Medical Oncology Department, Hospital Clinic and Translational Genomics and Targeted Therapies in Solid Tumors, IDIBAPS, 08036 Barcelona, Spain; 7Department of Pathology, Biomedical Diagnostic Center, Hospital Clínic of Barcelona, University of Barcelona, 08036 Barcelona, Spain; 8Neurological Tissue Bank of the Biobank, Institute of Biomedical Research August Pi i Sunyer (IDIBAPS), 08036 Barcelona, Spain; 9Laboratorios Gebro Pharma S.A., 08022 Barcelona, Spain; 10Department of Cell Biology, Physiology and Immunology, Faculty of Biology, University of Barcelona, 08028 Barcelona, Spain; 11Institute of Biomedicine, University of Barcelona (IBUB), 08036 Barcelona, Spain; 12Department of Medicine and Life Sciences (MELIS), Faculty of Health and Live Sciences, Universitat Pompeu Fabra, 08036 Barcelona, Spain

**Keywords:** 5-ALA, 3D tumor models, neurological cancers, drug screening, glioblastoma, organoids, personalized medicine, spheroids, photodynamic therapy

## Abstract

Background: The high recurrence of glioblastoma (GB) that occurs adjacent to the resection cavity within two years of diagnosis urges an improvement of therapies oriented to GB local control. Photodynamic therapy (PDT) has been proposed to cleanse infiltrating tumor cells from parenchyma to ameliorate short long-term progression-free survival. We examined 5-aminolevulinic acid (5-ALA)-mediated PDT effects as therapeutical treatment and determined optimal conditions for PDT efficacy without causing phototoxic injury to the normal brain tissue. Methods: We used a platform of Glioma Initiation Cells (GICs) infiltrating cerebral organoids with two different glioblastoma cells, GIC7 and PG88. We measured GICs-5-ALA uptake and PDT/5-ALA activity in dose-response curves and the efficacy of the treatment by measuring proliferative activity and apoptosis. Results: 5-ALA (50 and 100 µg/mL) was applied, and the release of protoporphyrin IX (*PpIX*) fluorescence measures demonstrated that the emission of *PpIX* increases progressively until its stabilization at 24 h. Moreover, decreased proliferation and increased apoptosis corroborated the effect of 5-ALA/PDT on cancer cells without altering normal cells. Conclusions: We provide evidence about the effectiveness of PDT to treat high proliferative GB cells in a complex in vitro system, which combines normal and cancer cells and is a useful tool to standardize new strategic therapies.

## 1. Introduction

Glioblastoma (GB) remains the most frequent primary brain tumor in adults, with an incidence of 4–5 cases per 100,000 persons per year and a median overall survival time of 15 months [1]. Less than 9.8% of patients with GB achieve 5-year survival, representing the lowest survival rate of malignant brain tumors [2]. The standard treatment is tumor resection followed by concomitant radiotherapy associated with Temozolomide and posterior sequential Temozolomide (STUPP-protocol) [3,4]. Ultimately, curative treatment is still lacking for GB despite all efforts that have been made.

One aspect to consider is that GB invariably recurs in a more aggressive form, and, importantly, more than 80% of the recurrences are located adjacent to the resection cavity. This phenomenon is thought to derive from micro-tumoral infiltration of the vessels and white matter tracts [5,6], which goes inadvertent during surgery, and from senescence-induced changes after radiotherapy [7,8]. Thus, to prevent tumor recurrence and improve patient survival rates, a plausible solution would be a locally-delivered therapy able to eliminate the microscopic disease left after surgical resection. In this sense, the photosensitizer 5-aminolevulinic acid (5-ALA) could become an appropriate aid [9,10].

Already, 5-ALA represents a standard alley for intraoperative fluorescence-guided oncologic neurosurgery, allowing visualization of the tumor margins and infiltrating regions. Additionally, the 5-ALA compound is a precursor to the potent photosensitizer, protoporphyrin IX (*PpIX*). When *PpIX* is excited by 635 nm light, it generates triplet oxygen species triggering cytotoxicity as a photosensitizer [11,12,13,14,15,16]. Therefore, if we use both the fluorophore and photosensitizer (PS) capabilities of protoporphyrin IX (*PpIX*) at the same time, we will be able to visually identify neoplastic tissue and, at the same time, we can selectively destroy that tissue, thus improving the GB treatment success. This synergistic mechanism of action, which integrates both diagnostic and therapeutic capabilities into a theragnostic single product, named photodynamic therapy (PDT), has provided exciting prospects for optimizing treatment outcomes in GB [17]. Finally, a major challenge of PDT is attaining an even delivery of the photostimulation over the irregular surgical cavity without causing thermal injury to the normal brain tissue. We have modeled human cerebral organoid-glioblastoma initiating cells (GICs) co-cultures to test different PDT strategies with 5-ALA against the tumor and their possible effect in the surrounding parenchyma. This model allows the evaluation of both the therapeutic efficacy and the eventual damage to normal cells.

## 2. Materials and Methods

### 2.1. Cell Cultures

#### 2.1.1. Tumorsphere Cultures

GICs from two different glioblastoma subtypes: proneural (GIC7) (from Dr. Marta María Alonso, Department of Pediatrics, Clínica Universidad de Navarra, University of Navarra, Pamplona, Spain) and mesenchymal (PG88), were obtained from human GB specimens as previously described and characterized by their different radio-sensibility [18,19,20]. Both types of GICs belong to non-mutated and non-G-CIMP subtypes and grow up as tumorspheres in non-laminin coated plates and in adhesion on 7.5 mg/mL laminin-coated plates (Sigma, St. Louis, MO, USA) [21]. Cells were maintained in a complete Neuronal Stem Cell (NSC) medium constituted by Dulbecco’s Modified Eagle Medium and Nutrient Mixture F-12, DMEM/F12, (Invitrogen Waltham, MA, USA; 11320-033;), containing 2 mM glutamine, supplemented with N_2_ (Gibco A1370701), 4.5% glucose (Sigma, Merck KGaA, Darmstadt, Germany), 1M Hepes (Sigma, St. Louis, MO, USA), 2% BSA (Sigma, St. Louis, MO, USA), 20 ng/mL FGF-2 (R&D Systems; 233-FB-25/CF), and 20 ng/mL EGF (R&D Systems; 236-EG-200), at 37 °C, in a humidified 5% CO_2_ and 5% O_2_ atmosphere (hypoxia conditions), to simulate brain microenvironment (Heracell 150i incubator). GICs were infected with lentiviral vectors carrying green fluorescent protein (GFP) to trace the tumor burden growing in each organoid [22].

#### 2.1.2. Cerebral Organoids

The human BJiPSC-SV4F-9 cell line was used to generate neurocortical brain organoids. For patterning of neurocortical spheroids, pluripotent BJiPSC-SV4F-9, colonies cultured on Geltrex (Invitrogen, Waltham, MA, USA; A15696-01) were lifted with TrypLE™ Express (Gibco, Carlsbad, CA, USA: 12605) at 37 °C for 5 min. Single cells were transferred to individual low adherence V-bottom 96-well plates (S-Bio Prime, Hudson, NY, USA; MS-9096VZ) in 200 µL of spheroid starter media with 10 µM Rock inhibitor Y-27632 (Calbiochem, San Diego, CA, USA: 688001), 10 µL dorsomorphin (Sigma, San Luis, MO, USA:PS499), and 10 µM SB-431542 (Sigma, San Luis, MO, USA; S4317). Spheroid starter media was DMEM/F12 (Invitrogen, Waltham, MA, USA; Invitrogen, Waltham, MA, USA; Invitrogen, Waltham, MA, USA; 11320-033), containing 20% KnockOut Serum (Invitrogen, Waltham, MA, USA; Invitrogen, Waltham, MA, USA; 1287-010), non-essential amino acids (Invitrogen, Waltham, MA, USA; 11140050), Glutamax (Invitrogen, Waltham, MA, USA; 35050061), β-mercaptoethanol, and 100 U/mL penicillin–streptomycin. The same media without a ROCK inhibitor was used for the next 5 days, after which the media was changed to Neurobasal-A-based spheroid media. Neurobasal-A spheroid media was composed of Neurobasal-A medium (Invitrogen, Waltham, MA, USA; 10888022) with B-27 serum substitute without vitamin A (Invitrogen, Waltham, MA, USA; 12587), Glutamax (Invitrogen, Waltham, MA, USA; 35050061), and 100 U/mL penicillin–streptomycin. From day 7 to day 25, 20 ng/mL FGF-2 (R&D Systems, McKinley Place NE, MN; 233-FB-25/CF) and 10 ng/mL EGF (R&D Systems, McKinley Place NE, MN; 236-EG-200) were added to the Neurobasal-A spheroid media and the spheroids were cultured in 96-well plates through day 25, with daily half-media changes. Fresh FGF-2 and EGF were added daily. The medium was renewed daily from day 7 to 15 and every other day from day 17 to 25. At day 20, spheroids were collected in a 15 mL tube (using a 10 mL strippette) and transferred to ultra-low attachment 6-well plates (Corning, New York, NY, USA; CLS3471) at a density of 8–10 spheroids per well and cultured through the remainder of the protocol in 2 mL of media per well. Neural differentiation was induced between days 27 and 41 by Neurobasal-A spheroid media supplemented with 20 ng/mL BDNF (R&D Systems, McKinley Place NE, MN, 248-BD) and 20  ng/mL NT-3 (R&D Systems, McKinley Place NE, MN, USA; 267-N). Half-volume media changes were performed every other day between days 27 and 41. From day 41 spheroids were maintained in Neurobasal-A spheroid media for the generation of assembled GICs from each cell line and were added either as single cells or as tumorspheres, 2000 cells, or 2 tumorspheres for each organoid. For each condition, 16 wells were used to obtain 16 replicates for each co-culture scenario, both GFP-GIC7 and GFP-PG88 as single cells or tumorspheres. The co-cultured was maintained for 20 days before cells were fixed, on different days, for immunofluorescence analysis.

### 2.2. Double Immunofluorescence (IF) Analysis

Whole organoids were fixed with 4% paraformaldehyde (PFA) for 20 min, washed with PBS prior to overnight incubation in 30% (weight/volume) sucrose at 4 °C, and embedded in OCT. Cryosections of 12 μm were obtained with Leica CM1950 cryostat (Leica Biosystems, Barcelona, Spain) and stored at −80 °C for long storage. Sections were then allowed to equilibrate at room temperature, encircled with the hydrophobic super PAN pen (Sigma-Aldrich, Z672548-1EA, Madrid, Spain), and dried for 5 min. The sections were permeabilized in 0.4% Triton X-100 rinsed and blocked in 5% goat serum at room temperature. Then, the sections were incubated overnight with primary antibodies: anti-Ki67 (Anti-Ki67 antibody ab16667, Abcam, UK) at 1/150 dilution, anti-TUJ1 (1:200; Sigma, T5076, Saint Louis, MO, USA) to study the neuron-specific class III beta-tubulin expression, and anti-*GFAP* (1:100; Dako, Z0334, Santa Clara, CA, USA) in a humidified chamber. The following secondary antibodies were used: Alexa Fluor 594 goat anti-mouse (1:250; Invitrogen, A11005, Bleiswijk, The Netherlands) and Alexa Fluor 647 goat anti-rabbit (1:200; Invitrogen, A21246, Bleiswijk, The Netherlands). Sections were mounted and nuclei stained in Fluoromount DAPI (SouthernBiotech, 0100-20, Birmingham, AL, USA). Images were captured at 10×, 20×, and 40× magnifications, using an Olympus BX41 (I.C.T, S.L., Barcelona, Spain) inverted fluorescence microscope. Images were analyzed using ImageJ.

To compare different areas of the tumor-organoid co-cultures, at least three immunofluorescence (IF) slides from replicate samples were evaluated and analyzed in at least three fields of each sample. Then, we evaluated the IF area of protein after normalizing the background. Results were expressed as the mean and standard deviation of protein expression values at each condition to be statistically compared.

### 2.3. TUNEL Analysis

Terminal deoxynucleotidyl transferase-mediated dUTP nick-end labeling (TUNEL) assay was determined to analyze apoptotic cells in tumor-organoid co-cultures. We used ApopTag In Situ Detection Kit (Merck Millipore, S7165, Darmstadt, Germany). Samples were obtained as described above. Tissue cryosections were air-dried at room temperature for 5 min, rinsed in PBS 1X, post-fixed, and permeabilized in precooled ethanol: acetic acid 2:1 for 5 min at −20 °C. After washing three times, sections were incubated briefly at room temperature with an equilibration buffer and then incubated for 1 h at 37 °C with TdT enzyme reaction mixture in a humidified chamber, which was stopped by stop/wash buffer for 10 min at room temperature, prior to incubating with anti-digoxigenin conjugate (rhodamine-labeled) for 30 min avoiding exposure to light.

Fluor mount DAPI was used as a nuclear counterstain. Images were taken with an Olympus BX41 (I.C.T, S.L., Barcelona, Spain) fluorescence microscope using standard fluorescein excitation and emission filters.

Images were analyzed using ImageJ v1.53to assess the percentage of dead cells. TUNEL-positive nuclei (pink/magenta) were counted at different fields of each sample and at least in three different slices from replicates, considering tumor areas and non-tumor areas of each field.

### 2.4. Photo Dynamic Therapy (PDT)

#### 2.4.1. 5-ALA Treatment and Viability Assays in Glioblastoma Cell Cultures

The viability under 5-ALA treatment and irradiation was assessed in GIC7 and PG88 cells, both as adherent cultures and growing as tumorspheres. For the cytotoxicity assessment at different concentrations in the two cell lines, cells were seeded in flat-bottomed laminin-coated 96-well plates at 10^4^ cells/well density. For the establishment of organoid co-cultures, we used tumorspheres in uncoated U-shaped plates to facilitate centrifugation steps of 5 min at 800 rpm, carried out to perform media replacements. Different plates were used for the non-irradiated cells, and different irradiation intensities were applied (0.6, 1.2, 2.4, and 4.8 J/cm^2^). After 24-h incubation (48 h in the case of tumorspheres), the media was removed, and new media with different concentrations of 5-ALA (100, 50, 25, 12.5, 6.25, and 3.75 µg/mL) prepared from the 5-ALA 8 mg/mL stock solution were added. Untreated cells were incubated in the same medium composition without 5-ALA. The 5-ALA concentrations and distribution throughout all the plates prepared were identical. After 24 h of incubation, the medium was renewed, and each plate was exposed for different times (0, 15, 30, 60, and 120 s) using a Suntest CPS (Atlas, USA) solar simulation unit equipped with a xenon arc lamp (200–800 nm). The irradiance was set to 400 mW/m^2^, resulting in the irradiation doses mentioned above (4.8, 2.4, 1.2, and 0.6 J/cm^2^). Plates were placed again in the incubator under standard conditions for 24 additional hours. Then, cell viability was assessed using the CytoTox96 non-radioactive cytotoxicity assay (Promega, San Francisco, CA, USA) under the manufacturer’s instructions. Four hours after the addition of the resazurin reagent (24 h in the case of neurospheres), fluorescence was measured using a fluorescence plate reader at 560/590 nm excitation/emission (Infinite M Plex, Tecan). The viability was represented as the percentage of fluorescence with respect to the untreated cells in each of the plates.

#### 2.4.2. 5-ALA Treatment and Viability Assays in Glioblastoma Cell Cultures

Kinetic assays to evaluate *PpIX* generation in tumorspheres were performed in 96-well plates. GIC7 and PG88 cells were seeded at 10^4^ cells/well in 100 µL of media and incubated for 48 h to leave cells to self-aggregate into tumorspheres. Then, plates were centrifuged for 5 min at 800 rpm, and the medium was replaced by 100 µL of medium containing 5-ALA at 100 and 50 µg/mL. The 5-ALA stock solution was prepared from the powder at 8 mg/mL using culture medium and NaOH 1M to neutralize the solution to pH 7 (specifically, 60 µL of NaOH 1M were added to a final volume of 1.25 mL for 10 mg of 5-ALA powder). The solution was sterilized using a 0.22 µm filter before use and kept frozen for further uses. Untreated cells were incubated using the same medium composition without 5-ALA. Six replicates were performed for each concentration. The metabolism of 5-ALA to *PpIX* was evaluated at different time points using a fluorescence plate reader at 405/635 nm excitation/emission (Infinite M Plex, Tecan Trading AG, Switzerland. Plates were taken from the incubator for fluorescence reading and placed back in the incubator until the next kinetic time point. The plate lid was not removed for the measurement to preserve the plate sterility, and the fluorescence intensity was measured from the bottom of the plate. For calculations, the fluorescence values from untreated cells were subtracted from the fluorescence values obtained at each time.

#### 2.4.3. 5-ALA/PDT Treatment of GFP-GICs Co-Cultures

The experiment was carried out in two twin 96-cell plates, one for irradiation at 1.2 J/cm^2^ and the other used as non-irradiated control. In each plate, there were at least six replicates of brain organoids alone, GIC7 and PG88 cells as tumorspheres alone, brain organoids co-cultured with GIC7, and brain organoids co-cultured with PG88.

The treatment with 5-ALA (50 µg/mL) or medium alone was added to the cell wells, and after 24 h, plates were irradiated or not at 1.2 J/cm^2^ as described above. At this point, *PpIX* and GFP fluorescence was measured (405/635, excitation/emission, and 485/530 excitation/emission, respectively) at 12 different points of each well to capture the fluorescence from the entire well surface. These measurements were done without removing the plate lid and from the bottom of the plate. Additionally, 25 µL of the cell media were taken to perform the lactate dehydrogenase (*LDH)* assay (Promega, San Francisco, CA, USA) to have a measure of cell death just after irradiation. Next, plates were incubated again for 24 additional hours, and the GFP fluorescence measurement was carried out again. This procedure was repeated at 72 h post-irradiation. To determine the effect of 5-ALA/PDT on the GICs-tumorspheres viability, the GFP ratio between the fluorescence endpoint measurements and the first measurement after irradiation was obtained. The global cell viability of each well was determined using the *LDH* assay, using the ratio between the *LDH* present in the medium 72 h post-irradiation and the amount in the same well just after irradiation (t = 0). Since the amount of *LDH* released to the medium correlates with cell death, increasing values of this ratio indicated cell death enhancement under the specific experimental conditions tested. Additionally, at 72 h post-irradiation, pictures were taken under the GFP filter using a high-resolution microscope EVOS M7000 (Invitrogen, Thermo Scientific, Waltham, MA, USA).

### 2.5. Statistical Analysis

Data are presented as mean ± SEM. Two-tailed Student’s *t*-test (to compare two experimental groups) or ANOVA (to compare three or more groups) were performed for data analysis using GraphPad Prism v.5.0 Software for Windows (GraphPad Software, San Diego, CA, USA). Bonferroni post-test was performed to compare replicate means. For all statistical methods, *p* < 0.05 was considered significant and was denoted as: * *p* < 0.05, ** *p* < 0.01, and *** *p* < 0.001.

## 3. Results

### 3.1. GICs Co-Cultured with Cerebral Organoids Generate a 3D Model of GB Tumor into the Normal Brain-like Structures

The pluripotent stem cell line BJiPSC-SV4F-9 cell line owned the intrinsic ability to self-organize into brain-like structures (so-called brain organoids) after subsequent 3D culture conditions [23] evolving from neuronal progenitors into neurons and glial cells at days 30 and 42, respectively (Figure 1A).

The multicellular aggregate derived from induced pluripotent stem cells can generate a group of polarized neural progenitors mimicking the developing cortex. The conformed brain organoids contained cells expressing neural, stem, and glial cell markers, as confirmed by immunofluorescence (Figure 2). Cell clusters at day 43 of culture expressed *TUBB3* (*TUJ1*), a marker of the early stage of differentiation of neuronal cells, and *GFAP*, compatible with astrocytes and radial glial cells. Moreover, the transcription factor *SOX2*, an embryonic stem-cell marker [24], was expressed homogeneously in the cell mass; the same occurred with the *O4* marker, which marks the oligodendrocytes [25]. These organoids were used to model GB invasion and infiltration by establishing GICs co-cultures of GIC7 and PG88 cells, previously infected with lentiviruses carrying green fluorescence protein (*GFP*), facilitating the identification of GFP-GICs co-cultured with organoids.

GFP-GICs were co-cultured with cortical brain organoids at day 43 of differentiation, representing a time-point displaying neurons and astrocytic cells (Figure 1A,B). Co-culture was performed by adding, for each GIC line, two tumorspheres or 2000 isolated GFP-GICs per organoid. These proportions induced successful engraftments in both GFP-GIC7 and GFP-PG88 cells (Figure 1B).

We microscopically monitored the tumor cells’ growth in the organoid at least 30 days of co-culture (DoC) (Figure 3A). The GFP-GICs in contact with the organoid started their engraftment 24 h after the setting up of the co-culture. On DoC 15, we observed that GFP-GIC7 and GFP-PG88 adhered to the organoid surface, infiltrating the organoid and growing on it (Figure 3A). On DoC 41, the organoids were analyzed by confocal microscopy to visualize the cell bulk of tumor infiltration (Figure 3B), confirming that both GFP-GICs efficiently invaded and proliferated into the brain organoid.

### 3.2. Phenotypic Characterization of GFP-GICs Co-Cultured with Cerebral Organoids Shows an Astrocyte Accumulation at the Scare Area

We followed the progression of GFP-GIC growth into organoids in co-cultures replicate (*n* = 3) from DoC 24 to 37 to check if any expression changes occurred in cell populations along with the tumor growth (Appendix A). Assembled replicates showed similar and homogenously distributed *TUJ1*^+^ neuronal cells (Appendix A). In contrast, GFAP^+^ astrocytes were progressively located closer around the tumor from DoC 24 to DoC 31 and DoC 37 (Appendix A) when we followed GPF-GICs-organoids growth. A significant progressive increase of *GFAP* protein expression from DoC 24 to 37 (*p* < 0.05) surrounding tumor masses concerning the expression on peripheral areas was observed (Appendix A). Moreover, additional morphological changes occurred on GFAP^+^ cells at DoCs 31 and 37, which changed from small and rounded to elongated, suggesting a glial infiltration around the tumor. Since the elongated *GFAP*^+^ cells around the GFP-GICs indicate reactive astrocytes, the process extension toward the injury site might mimic a glial scar [26] to protect the healthy tissue.

### 3.3. Tumor Engraftment Shows Effective Cell Surface Cell Growth as Well as Infiltration Capability of the GIC Cells into the Brain Organoids

We carried out co-cultures of GFP-GICs on differentiated brain organoids, either in the form of tumorspheres or as disaggregated GFP-GICs, for both GFP-GIC7 and GFP-PG88 cells. We did not find statistical differences between both GFP-GIC-organoid co-culture engraftment and growth either GICs were added as single cells or as tumorspheres (Figure 4). Moreover, both GFP-GICs invaded the organoid inside since both GFP-GICs were found in depth when consecutive organoid tissue slides were performed (Appendix A).

The striking ability of GFP-GICs to proliferate and infiltrate the cerebral organoid simulated the infiltrative phenotype of glioblastoma tumors. This cellular framework provided a potential mechanism to cell–cell signaling and intrinsic secretion of growth factors, guiding neuronal and glial development, altogether supplying a perfect microenvironment to GFP-GICs invasion infiltrating organoids.

### 3.4. Irradiation of 5-ALA Treated Glioblastoma Cells Induces Cytotoxicity

A kinetic study was performed to evaluate the production of *PpIX* after 5-ALA incubation in GIC7 and PG88 glioblastoma cells. Two different concentrations of 5-ALA (50 and 100 µg/mL) were applied to tumorspheres GIC7 and PG88 cell lines, and the release of *PpIX* fluorescence was measured at 1, 2, 4, 8, 24, and 30 h. The results indicate that in both cell lines, *PpIX* is produced already in the first hour after 5-ALA addition at all concentrations tested, and its production increases with time until 24 h. After this time, *PpIX* levels start to stabilize (Figure 5). However, clear differences can be observed between the two cell lines, especially at 50 µg/mL. GIC7 tumorspheres showed a higher capacity to metabolize 5-ALA into *PpIX* compared to PG88 tumorspheres (*p* < 0.005). However, at higher concentrations of 5-ALA (100 µg/mL), the production of *PpIX* was similar in both cell lines.

To analyze the effects of *PpIX* photoactivation, both GIC7 and PG88 adherent cells were treated with different concentrations of 5-ALA, ranging from 3 to 100 µg/mL, to better assess cell cytotoxicity. After 24 h, different irradiation doses were applied to the cultures, from 0.6 to 4.8 J/cm^2^. Figure 6 shows the cell viability assessed 24 h later in both GIC7 (Figure 6 top) and PG88 (Figure 6 bottom) cells. Very low cytotoxicity (between 10 to 20% of cell death) was observed under 5-ALA treatment at concentrations below 50 µg/mL. However, clear effects on cell viability were observed at 50 µg/mL, and these were irradiation dose-dependent. At this concentration, the 0.6, 1.2, and 2.4 J/cm^2^ irradiation doses reduced the viability to 53, 32, and 18%, respectively. The irradiation dose of 4.8 J/cm^2^ almost killed all the cells, similar to what was observed at 100 µg/mL 5-ALA at 1.2, 2.4, and 4.8 J/cm^2^ irradiation.

When the results of GIC7 and PG88 adherent cells were compared under 50 µg/mL of 5-ALA and 1.2 J/cm^2^ irradiation, a higher resistance of PG88 to the therapy was observed (Figure 7A). Under these experimental conditions, the GIC7 culture had 75% of cell mortality, whereas PG88 cells presented no effects on cell viability. A similar experiment using GICs tumorspheres showed more resistance of PG88 to the 5-ALA combined with irradiation treatment (*p* < 0.05) than GIC7 cells (*p* < 0.005), as shown in Figure 7B.

### 3.5. Photodynamic Therapy Induces Cell Death to Glioblastoma Cells Infiltrated in Cerebral Organoids

Next, the PDT was applied to the tumorsphere-organoid co-culture. The fluorescence emitted by the *GFP* protein of the GICs was used to assess the different viability of tumor and normal cells in the co-culture. In addition, the global cell viability was assessed by measuring the lactate dehydrogenase (*LDH*) release to the medium 72 h after irradiation. Brain organoids, GICs tumorspheres, and their co-cultures were treated using 50 µg/mL of 5-ALA for 24 h (Appendix A). Then, an irradiation dose of 1.2 J/cm^2^ was applied to one of the two twin plates to induce the photoactivation of *PpIX*; the other plate was used as a control. At this point, the production of *PpIX* was confirmed in each of the wells by fluorescence, and cell viability was then measured at 24 and 72 h post-irradiation to assess the efficacy of PDT-inducing tumor cell death.

As shown in Appendix A, there were no significant changes in cell viability at 24 h, as reported by the *GFP* fluorescence measurements. However, after 72 h, a clear decrease in *GFP* fluorescence was observed between the GIC7 cells alone (*p* = 0.005) and in those co-cultured with the brain organoids (*p* = 0.01) (Figure 8A). Additionally, a decreasing tendency of PG88 fluorescence alone and in co-culture was observed, although it was not statistically significant. The *LDH* assay corroborated these results. As shown in Figure 8B, the ratio of *LDH* release at 72 h with respect to time 0 was significantly enhanced for the GIC7 (*p* < 0.05) and PG88 (*p* < 0.05) cultures alone and in co-culture with the brain organoids, where the differences increased in both cells (*p* < 0.005). However, the release of *LDH* did not increase in the control organoid wells cultured alone under the same conditions, indicating that the PDT was selectively inducing cell death in glioblastoma cells and was not affecting the cell viability of the organoids.

The fluorescence pictures taken at 72 h post-irradiation confirmed these results (Appendix A). The intensity of GFP fluorescence was shown to decrease under 5-ALA treatment followed by irradiation.

In addition, we performed IF analysis of GIC-organoid co-cultures after treatment to evaluate cell death using the TUNEL technique (Figure 9). The number of apoptotic cells increased after PDT in invasive lesions with regard to areas not invaded by GICs. Then, the number of GFP-GICs decreased in both GIC7 (Figure 9A) and PG88 (Figure 9B).

We compared apoptosis of GICs invading organoids with apoptosis in organoid surrounding areas without GICs (Figure 9C,D). The percentage of apoptotic cells was similar in 5-ALA control GIC7-co-cultures than after PDT treatment, 21% vs 26%, respectively (*p* = 0.30) (Figure 9C, small cheese graphic to the left side of the big one); and in 5-ALA control PG88-co-cultures than after PDT treatment, 17% vs 19%, respectively (*p* = 0.41) (Figure 9D, small cheese graphic to the left side of the big one). Therefore, PDT per se was unable to induce apoptosis in normal brain-organoid cells.

Moreover, apoptosis increased in tumors from GICs invading organoids after PDT either GIC7, 17% vs 37% (*p* < 0.0001) (Figure 9C, small cheese graphic to the right side of the big one) or PG88, 18% vs 31% (*p* < 0.0001) (Figure 9D, small cheese graphic to the right side of the big one). These results suggested that apoptosis was the cell death mechanism secondary to 5-ALA PDT, involving mainly tumor cells that capture the photosensitizer, whereas brain-organoid cells were not affected.

On the other hand, *Ki67* was used to evaluate the proliferation of GICs into invaded organoids. Ki67 decreased in GIC7 (Figure 10A) and PG88 cells (Figure 10B) after PDT suggesting a decreased proliferation ratio. The analysis of GIC7 5-ALA control samples showed 59% of positive nuclei vs 24% of positive nuclei on PDT-treated GIC-7 co-cultures (*p* < 0.0001) (Figure 10C, small cheese graphic to the right side of the big one). PG88 5-ALA control co-cultures had 63% of positive *Ki67* cells with regard to PG88 PDT treated co-cultures that had 29% of positive *Ki67* nuclei (*p* = 0.010) (Figure 10D, small cheese graphic to the right side of the big one). Thus, from these experiments, we concluded that PDT might effectively treat both pro-neural and mesenchymal GB cells, limiting their proliferative activity without damaging normal cells.

## 4. Discussion

Curing patients with a GB will require a better understanding of neuroscience and brain tumor biology to prevent tumor recurrence and improve patient survival rates. We show a new accurate preclinical model to test plausible solutions based on locally-delivered therapy, able to eliminate the microscopic disease left after surgical resection.

GIC-organoid co-cultures recapitulate many aspects of the disease and can be used to assess the efficacy of local therapies for GB, in particular, PDT with 5-ALA, allowing the assessment of eventual collateral damage to the surrounding brain parenchyma. Moreover, testing the efficacy of local treatment in a human microenvironment provides outcomes that can be extrapolated more accurately to the clinical setting.

Based on the induced pluripotent stem cell line BJiPSC-SV4F-9, the organoid model recapitulates the human brain development from induced pluripotent stem cells to cells expressing mature central nervous system (CNS) markers. We show that organoids are self-organized and differentiated in vitro, exhibiting similar features in cell-type diversity as in the developing human brain. Furthermore, GICs co-cultured with human-brain organoids are a successful in vitro model to study glioblastoma diffuse invasion and proliferation, mimicking the human brain microenvironment and the crosstalk between a cancer cell and normal cerebral tissue. Indeed, this 3D co-culture system helps explore new glioblastoma treatments with elements that are not recapitulated using a standard 2D culture system.

An interesting observation in co-cultures is the increased population of GFAP^+^ cells around the tumor from DoC 24 to 37. This cellular migration might be attributed to differences in the tumor microenvironment as it has been postulated that astrocyte interactions with glioma cells are essential for glioma invasion [27,28,29]. Still, if astrocytes become outnumbered by glioma cells, they can no longer protect neurons from glioma cell glutamate secretion and excitotoxicity [27]. Indeed, our model provides an additional tool to analyze gap junctions in astrocytes, which might be a crucial target for manipulation in preventing glioma invasion since the difference in the invasiveness of GB can be partly attributed to the surrounding astrocytes [28].

Of particular interest would be to determine the ability of the tumor to respond to local therapies, which could become part of the neurosurgical resources [30,31,32,33,34]. Organoids could inform of both the antitumorigenic effect of these therapies and the potential collateral damage to the surrounding brain tissue. In this context, special attention has been aroused by photodynamic therapy [35,36,37,38,39,40].

In this context, 5-ALA PDT stands as a very convenient pro-therapy since the molecule is already a standard of care to guide surgical resection in GB. As a fluorophore, 5-ALA delineates macroscopically the tumor margins and maximizes the extent of resection [41,42,43]. After resection, the photosensitizer effect of 5-ALA can be elicited by stimulation at a higher wavelength and PDT administered in the same surgical act.

Specific photosensitizers like 5-ALA are able to absorb photons, generate fluorescence and trigger phototoxic processes that become detrimental for the tumor cells [44], based on the generation of reactive oxygen species (*ROS*), resulting in cancer cell death mainly due to the activation of apoptosis [45]. 5-ALA is a non-proteinogenic amino acid produced from glycine and succinyl-CoA in mitochondria by ALA synthase. It is a precursor of phototoxic protoporphyrin IX (PpIX) in the heme biosynthesis pathway [46,47] with a strong absorption band in the 380–420 nm spectrum, emitting red fluorescence at 635–704 nm. *PpIX* is reported to preferentially accumulate in cancer cells over non-tumor cells, although the precise mechanism underlying this is unknown [44,48,49,50].

Furthermore, the GICs-organoid co-culture is a system able to detect functional differences between GB phenotypes. Lower production of *PpIX* by PG88 under the same conditions could explain the higher resistance of these cells to the therapy.

Besides efficacy, safety evaluation is the other crucial aspect to assess in a prospective therapy. In our experiments, significant cell death did not occur after exposing the control organoids to PDT under the same conditions, indicating that the PDT was selectively inducing cell death in glioblastoma cells without affecting the cell viability of neural and glial cells surrounding the tumor. These results indicate that co-culturing GICs with brain organoids is a reproducible model to analyze PDT efficacy and other new strategic therapies to fight against GIC cells, including the therapeutic consequences over the tumor and normal cells.

## 5. Conclusions

We provide evidence that brain organoids co-cultured with GICs are a powerful cancer avatar to study the effectiveness of therapeutic approaches, modeling the symbiosis between normal and malignant cells and providing a real insight into the tumor-microenvironment interaction. According to our in vitro experiments, PDT seems to specifically target glioblastoma cells while sparing the surrounding tissue. This locally-derived therapy could be an effective adjuvant after surgical resection against microscopic invasion, although these results deserve further investigation in in vivo preclinical models.

## Figures and Tables

**Figure 1 cells-12-01125-f001:**
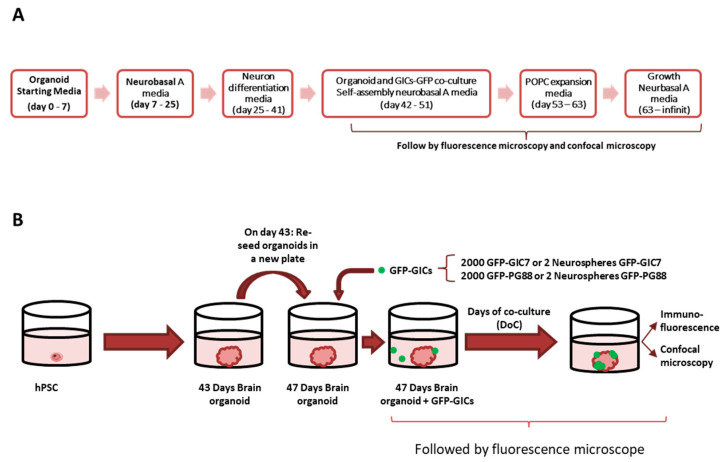
Schematic protocol of hiPSC cell cultures to induce neurocortical organoids and their co-culture with GFP-GICs. (**A**) Schematic protocol for neural induction of organoid cultures. The medium has been changed sequentially as indicated to obtain the different cell lineages. (**B**) On day 47, the GFP-GICs were seeded on top of the organoids to study their infiltration capacity on organoids. The obtained assembloid was followed by fluorescence microscopy until the end of the experiment when some of them were preserved for further histological analysis and confocal microscope imaging.

**Figure 2 cells-12-01125-f002:**
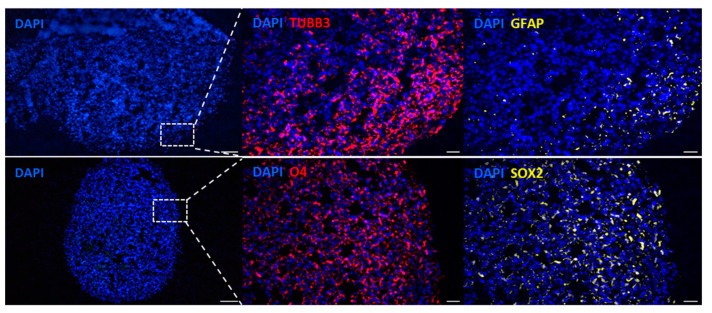
Immunofluorescence labeling of two differentiated organoids at day 43. The left panels show sections of organoids with blue nuclei stained with Fluoromount DAPI, scale Bar = 100 μm (10×). The dashed-line square represents the field visualized at higher magnification in the subsequent panels. The upper figures show the presence of neurons TUBB3 (red) and astrocytes GFAP (yellow). The bottom panels show oligodendrocytes O4 (red) and embryonic stem cells SOX2 (yellow). Scale bar 20 μm (40×).

**Figure 3 cells-12-01125-f003:**
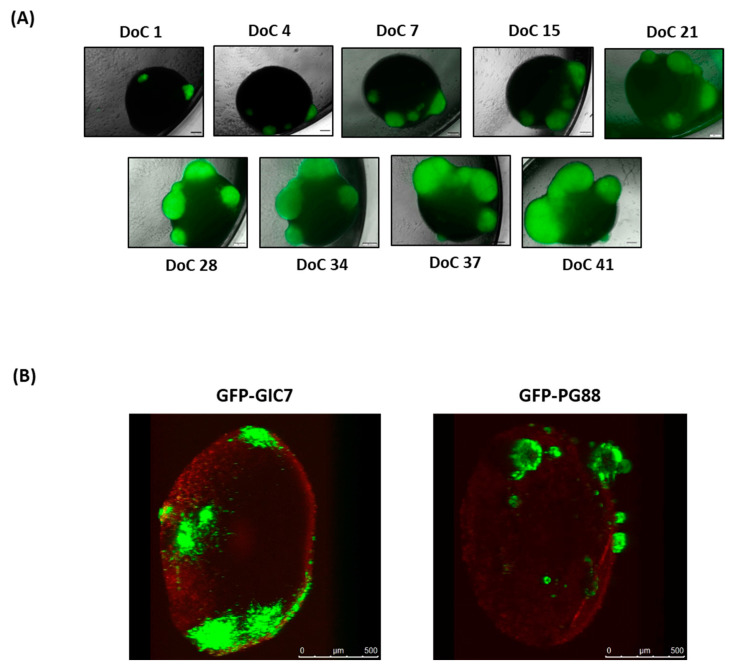
Follow-up of organoid-GICs co-culture by microscope. (**A**) Co-culture of GFP-GIC7 seeded as tumorspheres on top of the cerebral organoid, followed by fluorescence microscopy. Images were obtained from 1 to 41 days of co-culture (DoC) with a Microscopy Olympus IX51 at 4× magnification. Scale bar 200 μm. (**B**) Brain organoids were engrafted with 2000 single GFP-GICs proneural (left) and 2000 single GFP-mesenchymal (right) 24 days after seeding cells, which had already been differentiating for 47 days. Images were obtained with the confocal microscope. Scale bar 500 μm.

**Figure 4 cells-12-01125-f004:**
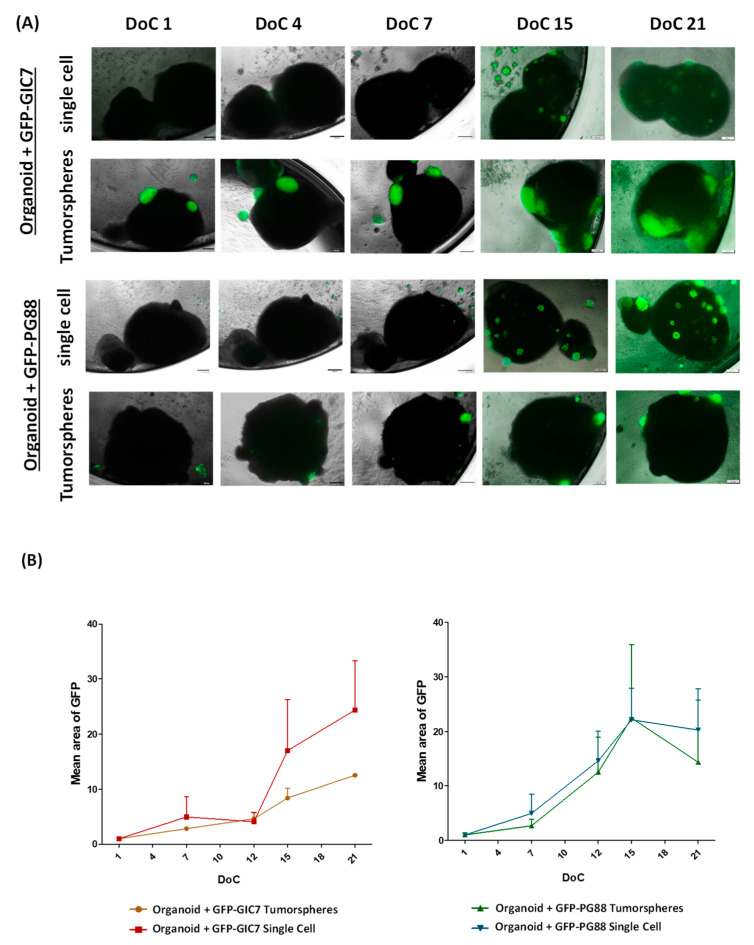
Study of the tumor engraftment effectiveness. (**A**) Merge of fluorescence and phase-contrast images of GFP-GIC-organoid co-cultures at days 1, 4, 7, 15, and 21 after the engraftment (DoC). Both GFP-GIC7 (top panels) and GFP-PG88 (bottom panels), seeded as tumorspheres or as a single cell, as indicated, achieve infiltration of the organoid generating a viable tumor. Both GFP-GICs grew around and inside the organoid. Images at 4× obtained with Microscope Olympus IX51 are shown. Scale bar 200 μm. (**B**) Semi-quantitative analyses of tumor growth expressed as the mean of tumor area (GFP positive area), concerning the total organoid, evaluating GFP fluorescence of GFP-GIC7 (left) and GFP-PG88 (right) seeded as tumorspheres or single cells. To normalize the data and homogenize the different co-cultures, the initial GFP fluorescence on DoC 1 was used to evaluate each well and to compare the GFP area. The data are representative of the tumor engraftment effectiveness of both GFP-GICs. An ANOVA test was used to compare the means of 4 different conditions for every DoC and each situation for every DoC vs. the first DoC. DoC = Days of co-culture.

**Figure 5 cells-12-01125-f005:**
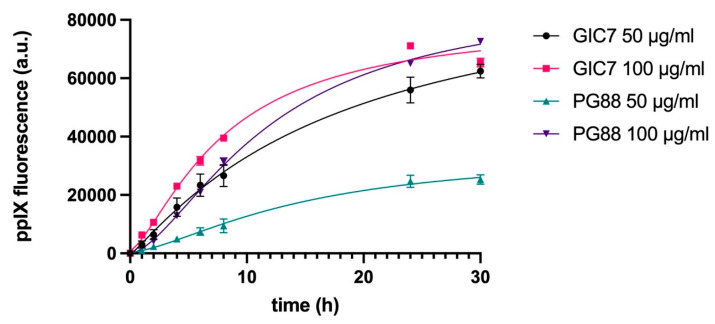
Kinetic study of *PpIX* emission of GICs. Both GIC7 and PG88 tumorspheres were treated at 50 and 100 µg/mL of 5-ALA and *PpIX* fluorescence was analyzed. Results show the mean ± SE of three independent experiments (*n* = 3) performed in triplicate and measured periodically until 30 h after 5-ALA administration.

**Figure 6 cells-12-01125-f006:**
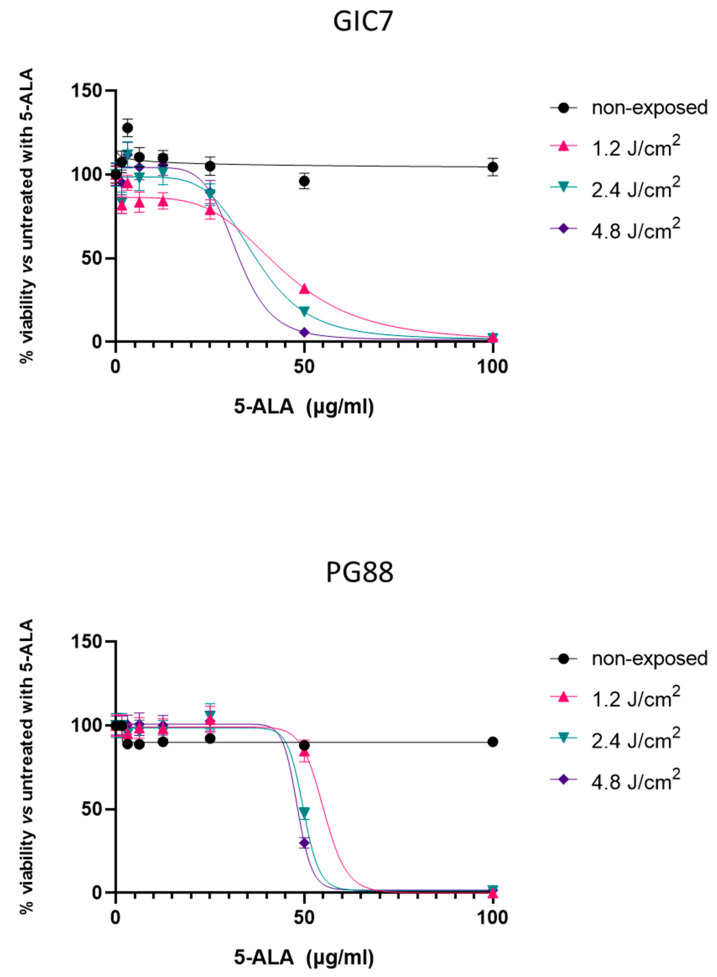
GIC7 (top) and PG88 (bottom) cell viability under different conditions of photodynamic therapy. The plot is representative of three independent experiments performed in both adherent cells in quadruplicate. Irradiation was applied at 1.2–4.8 J/cm^2^. The percentage of cell viability after 5-ALA treatment (0–100 µg/mL) was calculated with regard to control cells without treatment.

**Figure 7 cells-12-01125-f007:**
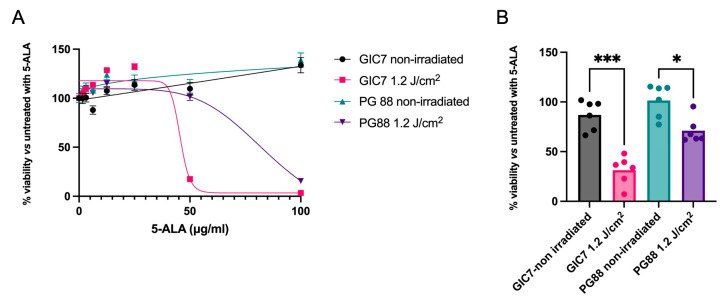
Comparison of GIC7 and PG88 cells’ behavior under 5-ALA treatment (0–100 µg/mL) and irradiation. (**A**) Dose-response comparison of adherent cell cultures with different 5-ALA concentrations treated with 1.2 J/cm^2^ irradiation. (**B**) Cell viability comparison between GIC7 and PG88 tumorspheres under 5-ALA 50 µg/mL and 1.2 J/cm^2^ irradiation. Results are representative of three independent experiments (*n* = 3) performed in triplicate. * *p* value < 0.05; *** *p* value < 0.005.

**Figure 8 cells-12-01125-f008:**
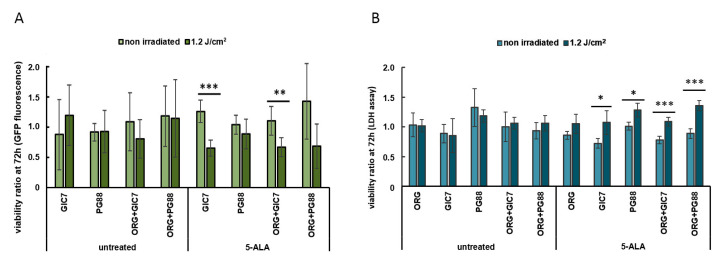
Cell viability after PDT of tumorspheres alone and co-cultured with brain organoids. (**A**) GICs viability after 72h post-irradiation using the GFP fluorescence ratio between each endpoint and time 0. (**B**) Global cell viability considering the ratio between the LDH release at 72 h post-irradiation and at time 0. Results are the mean ± SE of two independent experiments (*n* = 2) performed using six replicates. * *p* < 0.05, ** *p* < 0.01, *** *p* < 0.005.

**Figure 9 cells-12-01125-f009:**
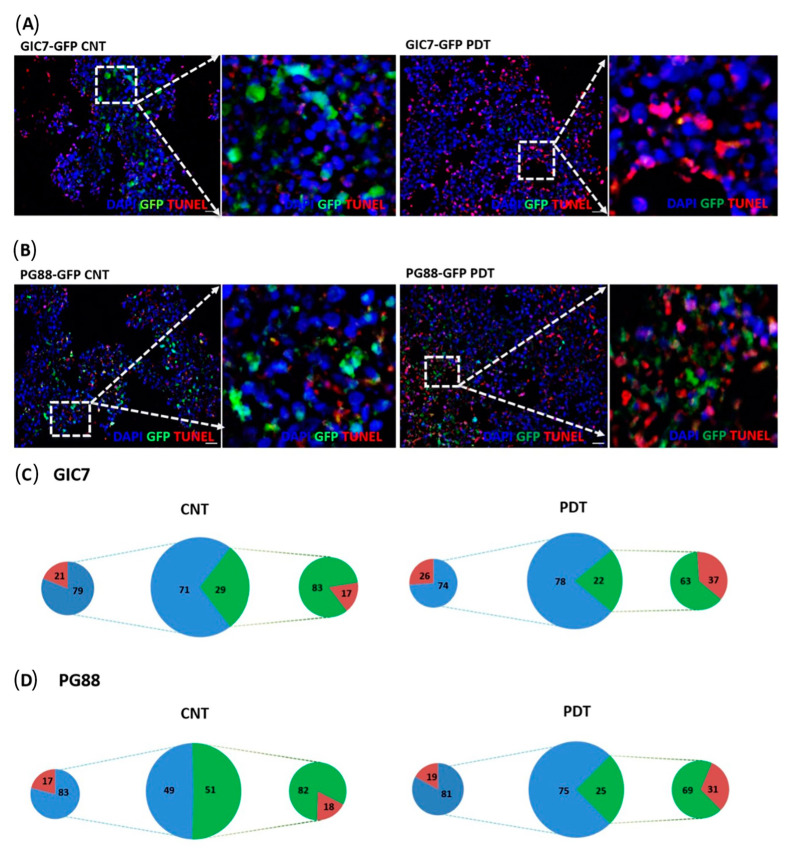
Tunel analysis of GFP-GICs and organoid co-culture. Tunel of GIC7-organoid co-cultures (**A**) and PG88-organoid co-cultures with 5-ALA (**B**) no treated (CNT) (left panels) or treated (right panels) with photodynamic therapy (50 µg/mL with 5-ALA, 1.2 J/cm^2^ irradiation). At 20× (big squares), the scale bar is 20 µm. 5-ALA controls (GIC7-GFP CNT or PG88-GFP CNT) without PDT and PDT-treated (GIC7-GFP PDT or PG88-GFP PDT) visualizing the red TdT-DIG-rhd cells and apoptotic bodies accumulated more in treated tissue at the tumor areas (green fluorescent cells) than in 5-ALA-controls (blue). Few apoptotic cells were scattered in non-infiltrated brain-organoid tissue (blue) in both 5-ALA-control and PDT-treated co-cultures. Blue DAPI-stained nuclei were used to counterstain cells. Semi-quantitative analysis of samples: GIC7 5-ALA CNT, 12 fields from 2 samples; GIC7 PDT-treated, 19 fields from 3 samples; PG88-5-ALA CNT, 7 fields from 2 samples and PG88 PDT-treated, 17 fields from 3 samples, was done. Semi-quantitative analysis is represented as a percentage average of positive cells on (**C**,**D**) pictures: green color (GFP-GICs cells), blue organoid cells, and red TdT-DIG-rhd positive cells; comparing controls (left part) and PDT samples (right part). Significant differences between GIC7-5-ALA CNT and GIC7-PDT-treatment (*p* = 0.0001) and between PG88-5-ALA CNT and PG88-PDT-treatment (*p* < 0.001) were found. (**C**) Cell counts show similar TdT-positive cells in organoid areas surrounding tumors (normal cells) and more TdT-positive cells in treated organoids than untreated samples (**D**).

**Figure 10 cells-12-01125-f010:**
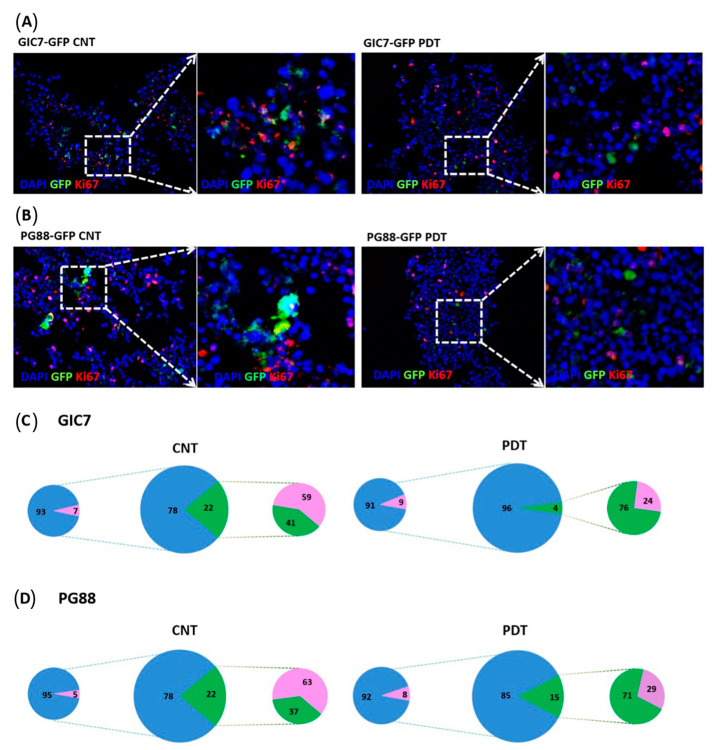
Proliferation study of GFP-GICs and organoids co-culture by Ki67. GIC7 and PG88-organoid co-cultures treated or not with PDT based on 5-ALA uptake after 5-ALA 50µg/mL treatment or medium alone. Fixed tissues were sliced at 12 µm to perform IF analysis with anti-Ki67 antibody (pink staining), scale bars 20 µm. Green fluorescent GICs infiltrated the organoid (green color), GIC7 (**A**) and PG88 (**B**) 5-ALA controls (left panels) show less positive pink nuclei (amplified in the small square at right) than PDT treated co-cultures (besides and amplified at right panels). Blue DAPI-stained nuclei were used to counterstain cells. Semiquantitative analysis of Ki67 positive cells was performed from: GIC7-5-ALA CNT, 7 fields from 1 sample; GIC7-PDT treated, 13 fields from 2 samples; PG88-5-ALA CNT, 8 fields from 1 sample and PG88-PDT treated, 11 fields from 2 samples. Results as a percentage average of positive cells are represented in (**C**,**D**) pictures: green color (GFP-GICs cells), blue DAPI-stained nuclei, and pink Ki67 positive cells; comparing controls (left part) and PDT samples (right part). Significant differences between GIC7-5-ALA CNT and GIC7-PDT-treatment (*p* = 0.0001) and between PG88-5-ALA CNT and PG88-PDT-treatment (*p* < 0.01) were found.

## Data Availability

Not applicable.

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
