# Peer review of "Preclinical Studies with Glioblastoma Brain Organoid Co-Cultures Show Efficient 5-ALA Photodynamic Therapy"

_cells, 2023, doi:10.3390/cells12081125_

Round 1

Reviewer 1 Report

This manuscript examines the use of 5-ALA as a potential therapeutic angle for glioblastoma using pre-clinical in vitro organoid models. There is some merit to this research however several concerns/comments need to be addressed as outlined below:

1. The method 2.1.2. mentions inhibitors including SB431542 etc. Thisis not reflected in the results outlined, or is this the routine culturing method used for these cells?

2.It is unclear why adherent cells were used for Fig 4...what does this add to the manuscript.

3. Were expts performed where treatment was given to glioblastoma organoids only without co-culture brain organoids? It would be important to evaluate whether the brain organoids are able to assist/or provide a signal to the glioblastoma spheroids to increase resistance to the therapies.

4. Some of the headings need to be more descriptive including the title.

 5. Use superscript for cell number 10not 104 cells.

Reviewer 2 Report

The work from Pedrosa et al. evaluates the efficacy of Photodynamic Therapy (PDT) to specifically target glioblastoma (GB) cells in brain organoids co-cultured with glioma-initiating cells. This is a fascinating study since it presents evidence of the potentiality of PDT to eliminate tumoral cells without affecting normal cells. This is a crucial feature that promotes PDT use as the standard treatment for glioblastoma-diagnosed patients since it demonstrates the non-tumoral cell safety of the therapeutic approach. The study design is correct, the results are well presented, and the conclusions are clear. However, in order to publish, several issues should be addressed.

1. In the introduction section, the PDT mechanism of action should be explained to introduce the technique better. Some exciting reviews regarding PDT for GB have been recently published. 

2. The scheme for organoid generation and co-culture assembly (Supplementary figure 1) should be presented as a figure in the manuscript since it clarifies the study.

3. In the results section, Figure 1 should be improved, particularly in the zoomed areas, since the resolution is not so good.

4. The authors performed a stack section of organoids-GICs co-cultures? For example, figure 2B could be improved by adding stacks.

5. Is there a statistical difference between GIC7 tumorspheres and single-cell co-cultures in graph 3B at day 21?

6. Why was the kinetic study of PpIX emission of GICs performed only in tumorspheres?

7. Line 350 reads “…. ranging from 3 to 200 ug/mL” …it should be 3 to 100 ug/mL? Please correct. 

8. Why the authors only performed GIC PDT using tumorspheres from GIC7 and not from PG88? Figure 5. 

9. The results presented in Figure 6 suggest that adherent mesenchymal GICs (PG88) are more resistant to PDT than CIC7, using a protocol combining 5-ALA [50ug/mL] and 1.2 J/cm2, which triggers ~50% reduction in cell viability on GCI7 tumorspheres. Since the determination in PG88 was not performed (or presented), why the authors used the above-mentioned experimental conditions on PG88 cells? Moreover, the PpIX Kinect study shows that using 50 ug/mL of ALA produces less PpIX in these cells, which could explain this result. Please clarify.    

10. Figure 7. Please correct legends B and C for A and B.

11. Line 526. There is an extra point. 

12. Supplementary Figure S3 should also present evidence of GIC7 cell infiltration into brain organoids.  

Round 2

Reviewer 1 Report

The authors have made suitable and acceptable changes/modifications to the manuscript based on my original comments. Therefore, I am happy for this manuscript to be accepted in current form.

Reviewer 2 Report

The authors answered all the points initially requested in the first revision round. 

Minor points.

Figure 1. The text is blurred; please provide a better-quality image in the final version of the manuscript. 

I recommend changing Figure 4B for the Reviewer Figure R1 since bars are more representative of the variability in the GFP area. 

Please specify in the figure legends which statistical analysis was performed.